# The Influence of OAT1 Density and Functionality on Indoxyl Sulfate Transport in the Human Proximal Tubule: An Integrated Computational and In Vitro Study

**DOI:** 10.3390/toxins13100674

**Published:** 2021-09-22

**Authors:** Jasia King, Silvia M. Mihaila, Sabbir Ahmed, Roman Truckenmüller, Stefan Giselbrecht, Rosalinde Masereeuw, Aurélie Carlier

**Affiliations:** 1MERLN Institute for Technology-Inspired Regenerative Medicine, Maastricht University, Universiteitssingel 40, 6229 ER Maastricht, The Netherlands; j.king@maastrichtuniversity.nl (J.K.); r.truckenmuller@maastrichtuniversity.nl (R.T.); s.giselbrecht@maastrichtuniversity.nl (S.G.); 2Division of Pharmacology, Utrecht Institute for Pharmaceutical Sciences, Universiteitsweg 99, 3584 CG Utrecht, The Netherlands; s.mihaila@uu.nl (S.M.M.); s.ahmed@uu.nl (S.A.); r.masereeuw@uu.nl (R.M.)

**Keywords:** indoxyl sulfate, organic anionic transporter, uremic toxins, albumin conformational changes, kinetic modeling

## Abstract

Research has shown that traditional dialysis is an insufficient long-term therapy for patients suffering from end-stage kidney disease due to the high retention of uremic toxins in the blood as a result of the absence of the active transport functionality of the proximal tubule (PT). The PT’s function is defined by the epithelial membrane transporters, which have an integral role in toxin clearance. However, the intricate PT transporter–toxin interactions are not fully explored, and it is challenging to decouple their effects in toxin removal in vitro. Computational models are necessary to unravel and quantify the toxin–transporter interactions and develop an alternative therapy to dialysis. This includes the bioartificial kidney, where the hollow dialysis fibers are covered with kidney epithelial cells. In this integrated experimental–computational study, we developed a PT computational model that focuses on indoxyl sulfate (IS) transport by organic anionic transporter 1 (OAT1), capturing the transporter density in detail along the basolateral cell membrane as well as the activity of the transporter and the inward boundary flux. The unknown parameter values of the OAT1 density $\left(1.15\times 10^{7}\;\mathrm{transporters}\;µm^{-2}\right)$, IS uptake ($1.75\times 10^{-5}\;µM^{-1}{\;s}^{-1})$, and dissociation ($4.18\times 10^{-4}{\;s}^{-1})$ were fitted and validated with experimental LC-MS/MS time-series data of the IS concentration. The computational model was expanded to incorporate albumin conformational changes present in uremic patients. The results suggest that IS removal in the physiological model was influenced mainly by transporter density and IS dissociation rate from OAT1 and not by the initial albumin concentration. While in uremic conditions considering albumin conformational changes, the rate-limiting factors were the transporter density and IS uptake rate, which were followed closely by the albumin-binding rate and IS dissociation rate. In summary, the results of this study provide an exciting avenue to help understand the toxin–transporter complexities in the PT and make better-informed decisions on bioartificial kidney designs and the underlining transporter-related issues in uremic patients.

## 1. Introduction

Approximately 10% of the global population is affected by end-stage kidney disease (ESKD). Unfortunately, worldwide, only 2.5 million people annually will receive renal replacement therapy [1,2], such as hemodialysis, to clear uremic toxins in the blood. According to the European Union Toxin (EUTox) database, a uremic toxin is a food and drug organic metabolite in the bloodstream that needs to be removed from the blood circulation. However, if these uremic toxins remain in the bloodstream, the patient will suffer from a number of complications, including, but not limited to, cardiovascular disease, anemia, and progressive kidney failure [3,4,5]. Uremic toxins are subdivided into three main categories according to their size and binding ability: (1) water-soluble (and non-protein bound; MW < 500 Da); (2) middle molecule (MW > 500 Da); (3) protein-bound uremic toxins (PBUTs) (MW < 500 Da). Hemodialysis falls short as a treatment since it only reproduces the glomerulus’ filtration function in removing water-soluble, some middle molecules, and not yet bound uremic toxins. The latter are not effectively cleared since they are too big to be removed via ultrafiltration once bound to albumin (MW > 65 kDa). Indeed, in normal physiological conditions, PBUTs are removed by active transport in the proximal tubule (PT). Therefore, to improve the effectiveness of hemodialysis, the functionality of the proximal tubule also needs to be mimicked.

Significant experimental strides have been made toward fabricating a bioartificial kidney (BAK) using hemodialysis hollow fibers populated with PT cells, e.g., genetically modified conditional immortalized proximal tubule cells expressing organic anionic transporter-1 (ciPTECs-OAT1) [6,7]. More specifically, PT cells create a solute barrier via the formation of tight junctions between adjacent cells. PT cells are well coordinated to be uniformly polarized along their basolateral (blood capillary side) and apical (urine/PT luminal side) membranes. In addition, both membrane surfaces have specific transporters to move unwanted solutes against the concentration gradient. These transporters include anti-porters, co-transporters, efflux pumps, and exchangers [8]. Considering the multitude of transporters and solute interactions and non-obvious kinetic influences on toxin clearance, computational models can support the research and design improvements to BAKs by providing a quantitative framework to model the combinatorial solute–transporter interactions. Moreover, the research performed to optimize BAKs can be accelerated at a reduced experimental cost by coupling experimental and computational models.

Computational models have been used throughout organ-on-chip and pharmacokinetic research and applied to understand physiological nephron function, including the influence of osmosis-driven and gradient-driven water reabsorption in the proximal tubule [9]; the effect of tubuloglomerular feedback dynamics on solute clearance by coupling a single nephron model [10]; and the effect of apical flow on water-soluble transport in single and coupled nephrons [10]. Other computational models have focused on replicating the transporter membrane interactions that influence the clearance of uremic toxins and solutes. For example, Layton used the available parameters from rat renal physiological models and made appropriate scaling-up assumptions for the transporter numbers in human models to investigate the effect of flow on transporter density [11]. Refoyo et al. developed a computational model to replicate an experimental model of the function of ciPTECs-OAT1cultured on a dialysis hollow fiber with constant indoxyl sulfate perfusion [12]. They were able to validate their model by matching their clearance rates with experiments [6]. However, all the models mentioned above lumped the transporter function and the transporter density together into a general inward flux. This assumption leads to a reduction in information on the influence and importance of the transporters on uremic toxin clearance and limits the use of models to investigate the combinatorial solute–transporter interactions.

Building on these modeling works, in this integrated experimental–computational study, we developed a proximal tubule computational model focusing on the organic anionic transporter 1 (OAT1) transporting indoxyl sulfate (IS, an anionic protein-bound uremic toxin), capturing in detail the transporter density along the basolateral cell membrane as well as the activity of the transporter. The model’s unknown parameters (i.e., the transporter density, transporter uptake, and dissociation rates) were determined through dedicated experiments of indoxyl sulfate uptake of ciPTECs-OAT1 monolayers in tissue culture well plates. A more physiologically relevant model was developed by including albumin interaction with IS in both healthy and uremic conditions (including and excluding albumin conformational changes). As such, the proposed computational model provides an exciting avenue to help understand the toxin–transporter intricacies in the PT and make better-informed decisions on BAK designs.

## 2. Results

### 2.1. Fitted Results Using 25 µM Indoxyl Sulfate (IS)

The computational model was fitted accurately and was validated with the experimental results of IS uptake by the ciPTECs-OAT1 monolayer (detailed in methods Section 5.1). The experimental time-series data of intracellular IS with an initial concentration of 25 µM ($IS_{t=0}=25\;µM)$ was converted into an average concentration at each time point (*n* = 6) and used as input data for the Parameter Fitting Model. After running the evolutionary programming solver in COPASI, the three selected parameters ($kf_{Uptake}$ and $kf_{dissociation}$ and $OAT1_{t=0}$) were fitted, as shown in Table 1.

New simulations were run with $IS_{t=0}=25,\;50,\;100\;\mathrm{and}\;500\;µM,$ using the fitted parameters of $kf_{Uptake}$, $kf_{dissociation}$, and $OAT1_{t=0}$ to validate the fitting results. The simulations results of the intracellular IS concentration ($IS_{Cell}$) were validated against the corresponding experimental time-series data (Figure 1). The simulated curves matched well with the experimental data points (Figure 1a,c).

#### Selection of IS Uptake Boundary Condition Function

Using $kf_{Uptake}$ as a constant (case 1) was the most reliable version of the uptake boundary condition, which was followed closely by modeling $J_{Uptake}$ as the Michalis–Menten function (case 3). All four uptake boundary condition cases (Equations (5)–(8)) were fitted with the average time-series data with 25 µM initial concentration of IS ($IS_{t=0}=25\;µM)$. The resulting parameter values are shown in Table 1. Case 2 was not able to be fitted to the data and was discarded from further investigations. Figure 1a shows that the fitted $kf_{Uptake}$ as a constant (case 1) was the most optimal fit resulting in the lowest RMSE for each time point (Figure 1d), which was closely followed by case 3 (Figure 1b). All cases, particularly case 4 (Figure 1c), had difficulty fitting 500 µM, giving rise to the largest RMSE. The difficulty of fitting 500 µM was likely due to the oversaturation of the OAT1 by the high IS concentration levels. IS binding with OAT1 has an IC50 value of 25 ± 4 µM with fluorescein [6], alluding to higher IS concentrations inhibiting OAT1.

The fitted parameters make sense in relation to the experimental results. The OAT1 density (1.15 × 10^7^ transporters µm^−2^ = 3.7 × 10^14^ molecules) was high when compared to the experimentally determined OAT1 in the cortical region of the kidney of 100 pmol/g (3.6 × 10^12^ molecules) [13]. These results were within two orders of magnitude of each other, and it noted that ciPTECs-OAT1 is genetically modified to overexpress OAT1, which may lead to overestimating the uptake step compared to in vivo samples.

### 2.2. Standard Model Simulations

The standard model allows investigation of the individual influence of the IS uptake rate by OAT1 ($kf_{Uptake})$ and the dissociation rate of IS into the cell monolayer $\left(kf_{dissociation}\right)$ and the initial concentration of OAT1 on the basolateral cell membrane ($OAT1_{t=0}$) (detailed in methods Section 5.4). With the parameter-fitting model calibrated and validated with the experimental time-series data, we used the calibrated model parameters $(kf_{Uptake}=1.75\times 10^{-5}{\;s}^{-1}{\;µM}^{-1},\;kf_{dissociation}=4.18\times 10^{-4}\;s^{-1}\;\mathrm{and}\;OAT1=1.15\times 10^{7}\;\mathrm{trasnporters}\;µm^{-2},$ see the results of Table 1: Case 1) in the standard model to explore the influence of the presence of 1 mM albumin, similar to other ciPTECs–OAT1 experiments [6,7]. Considering that we were modeling a closed system, we checked for mass conservation (see more details in the Appendix A Appendix A).

The simulations were extended over 17.5 h (until complete removal of IS) to display the complex concentration (IS bound to albumin) in the blood compartment. The intracellular IS concentration ($IS_{Cell}$) and the cleared IS concentration in the dialysate ($IS_{D}$) were plotted in Figure 2a. IS in the blood compartment binds immediately to albumin to form a complex that subsequently binds with OAT1 for transport across the cell membrane. The dynamics were as expected, with albumin having full recovery in the blood compartment as the complex binds to the OAT1 and intracellular IS reducing at a similar rate as IS increasing in the dialysate.

The sensitivity analysis was used to investigate the relationships between the standard model parameters and their ability to clear IS to the dialysate (Figure 2b) with a 20% decrease of each model parameter value. The results clearly show that the initial albumin concentration and binding to IS in the standard model have no effect on the IS transported to the dialysate compartment (sensitivity = 0%) and that the dissociation rate has a very low sensitivity (sensitivity = 0.93%). Interestingly, the transporter density (sensitivity = 5.84%) and IS uptake rate (sensitivity = 4.34%) had comparable sensitivities and would influence the IS clearance similarly.

We next also sought to explore larger changes in the individual parameters and their influence on IS transport. Figure 2c–f display 100-fold step changes for the specified parameter values. Figure 2c shows that increasing the initial albumin concentration above 1 mM results in no change in the IS clearance, indicating that the standard model used excess albumin. Likewise, decreasing the albumin concentration to 1 × 10^−2^ mM results in a small reduction in the rate of IS removal from the system. As previously mentioned in the sensitivity analysis, the transporter density (Figure 2d) and IS uptake (Figure 2f) result in similar IS transport influences. With a 100-fold decrease in the parameter values, all the simulations resulted in nearly zero IS removed from the system.

### 2.3. Physiological vs. Uremic Conditions

According to the EUTox database, the plasma levels of IS measured in a patient were around 2.5 µM (≈0.54 µg/mL) in physiological conditions and 180 µM (≈37.07 µg/mL) in uremic conditions. Applying these initial conditions to the standard model parameters (Table 1: Case 1) resulted in the dynamic profiles shown in Figure 3a,b.

In Figure 3c, similar to Figure 2b, albumin does not influence the removal of IS at the physiological or uremic IS concentrations ($IS_{B,t=0}=$180 µM). The most striking differences between the conditions occurred with the transporter density (sensitivity = 87.88%) and dissociation rate of IS from the transporter (sensitivity = 75.18%), where they were seen as the most sensitive parameters and the limiting factor in the uremic condition. However, the uptake rate in uremic concentrations was more sensitive than the physiological concentration, even if the albumin was in excess compared to IS initial concentration.

In summary, by changing the standard model to include uremic ($IS_{B,t=0}$ = 180 µM) and physiological ($IS_{B,t=0}$ = 2.5 µM) IS concentrations, we found increased sensitivities for transporter density, IS dissociation rate, and uptake rate under uremic conditions compared to physiological conditions. Both conditions had the same sensitivities for albumin binding and albumin concentration (sensitivity = 0%).

### 2.4. Albumin Conformational Changes Effect on Binding

Although the previous section indicated that an increase in IS concentration, as seen in uremic conditions, affected the IS dynamics, we next sought to explore the influence of albumin conformational changes on IS transport (detailed in Section 5.6).

The uremic condition in a kidney patient is complicated. It was reported that albumin might undergo conformational changes under uremic conditions, resulting in a difficult molecule to bind, but once IS binds, the albumin–IS complex was much stronger than in the physiological state [7,14]. Van der Made et al. [7] reported previously that the clearance of IS was 85% less effective using uremic versus physiological albumin. To replicate the conformational changes of albumin in uremic patients, the binding coefficients of IS to albumin ($Kf_{Complex})$ and uptake rate ($kf_{Uptake})$ were multiplied by a reducing factor f1 and f2, respectively.

A parameter scan was performed for f1 and f2 to find the best-fit pair of parameters to represent the conformational changes seen in uremic albumin. Figure 4a displays a heat map of all the f1 and f2 parameter sets scanned, the resulting grid plots the concentration of IS removed by the OAT1 ($IS_{D}$). The colors range from 130 µM (blue) to 0 µM (white), with 19.35 µM as orange. Limiting the total IS removed range to 19.35 ± 1.5 µM (85% reduction from the results with no conformational changes) makes it easier to distinguish which parameter set on the heat map resulted in the desired outcome (Figure 4b). Interestingly, there were limited f1–f2 pairs that resulted in only 19.35±1.5 µM IS removed, with no pair in the middle region of the heat map. Figure 4c shows the eight parameter sets that were close to 19.35 µM, with f1 = 1 × 10^−5^ and f2 = 0.044 being the most optimal parameter set to represent the conformational changes of albumin in the uremic condition simulations.

We next simulated IS removal in uremic conditions with these conformational change parameters. The simulation predicted that the ciPTECs-OAT1 monolayer would require more than 165 h (6.9 days) to completely remove 180 µM of IS in the uremic condition (Figure 5a).

A sensitivity analysis was performed on the individual parameters of the uremic condition model ($IS_{B,t=0}=180\;µM;$ f1 = 1 × 10^−5^ and f2 = 0.044) by decreasing the parameter values by 20% (Figure 5b). IS removal was found to be least sensitive to changes in albumin concentration (sensitivity = 6.63%). In uremic conditions with conformational changes, transporter density (sensitivity = 93.91%) and IS uptake rate (sensitivity = 82.65%) were the most sensitive and limiting factors in the simulation, followed by albumin binding (sensitivity = 59.21%). It was observed that once conformational changes (f1 = 1 × 10^−5^ and f2 = 0.044) in albumin were introduced to the uremic condition ($IS_{B,t=0}=180\;µM$) albumin binding and concentration become more important parameters for IS removal than in the scenarios with healthy albumin.

## 3. Discussion

It is a non-trivial task to design an effective renal replacement therapy with functional PT cell cultures. However, understanding the intricate transporter–toxin interactions, quantifying the functional transport density, and replicating the conformational changes of carrier proteins, such as albumin in the system, can guide researchers as to the essential parameters that can increase toxin removal from a uremic patient. In this study, we developed a computational model that defines the basolateral cell membrane of the PT as an inward boundary flux that incorporates the transporter density, transporter uptake, and dissociation rate as main parameters for indoxyl sulfate (IS) removal. The computational model was fitted and validated with reliable experimental time-series data of IS uptake in a well plate.

We found that the computational model can accurately replicate the OAT1 function and IS transport by modeling the IS uptake boundary condition as mass action kinetics using $kf_{Uptake}$ as a constant, where case 1 was the most reliable function of the uptake boundary condition, which was followed closely by modeling $kf_{Uptake}$ as the Michalis–Menten function (case 3). The model parameters operated at the saturation level, which can be seen in Figure 2c–f, where increasing the parameters has minimal effect on the IS removal. The transporter density is higher than what was reported for other cell lines [15], but this is a reasonable value, since the model represents uptake by the genetically modified cell line that overexpresses OAT1 [16].

The advantage of implementing the boundary condition that decouples the OAT1 density and function ($kf_{Uptake}$, $kf_{dissociation}$) was that we could investigate the transporter parameters’ sensitivities on the removal of IS in various conditions. It was seen that the standard model is influenced mainly by the uptake rate of IS from the transporter. It was evident that the standard model operates in excess albumin from Figure 3c, where there was 0% sensitivity and no effects on IS removal with a significant change in the albumin concentration. These results suggest there was sufficient albumin in a healthy and uremic patient, such that the albumin concentration (without conformational changes) was not a limiting factor for IS removal.

When investigating the physiological and uremic conditions, it was evident that they operate in excess albumin and transporter density. The albumin concentration and binding were similar in the physiological condition (2.5 µM) and uremic concentration (180 µM). The models were functioning at transporter density saturation, and increasing the transporter density beyond 1.15 × 10^7^ transporters µm^−2^ had a minimal effect on IS removal. The notable difference between the physiological and uremic concentrations was the time taken to remove IS entirely from the system (17.5 and 35 h, respectively). The uptake rate and transporter density equally limit the removal of IS in the physiological condition. In contrast, the transporter density and dissociation rate dominated the uremic condition without conformational changes, while the transporter density, uptake rate, and albumin binding rate dominated the uremic condition with conformational changes.

Research has shown that albumin undergoes conformational changes, mainly glycation, which altered the IS binding characteristics [7,17,18,19] and reduced its removal from the system by 85% [7]. The Uremic model captured these conformational changes using a reduction factor on the association function of IS to albumin ($f1\times Kf_{\;complex}$) and the IS uptake rate ($f2\times Kf_{\;Uptake})$. Transporter density appeared to be the least sensitive parameter, and research efforts should not be put toward increasing the transporter density if there were more than 1.15 × 10^7^ transporters µm^−2^ present along the cell membrane. To understand the best approach in optimizing IS removal in uremic patients, the above results implied that experimentalists should focus on (1) investigating the role of albumin in toxin removal during dialysis sessions; (2) investigating physical, electrical, or chemical stimuli for increasing IS dissociation rate from the OAT1 [20]; and (3) quantifying the functional transporter density of the basolateral epithelial membrane.

Although the model was calibrated using experimental data, there were a few notable limitations of the model. First, the experiments were performed with a genetically modified cell line (ciPTECs-OAT1). Future experiments should include isolated proximal tubule cells to investigate transporter density variability among donor patients. Second, the model only replicates the interaction with IS and OAT1, which was a grand oversimplification of the in vitro and in vivo situation where there are over 400 genetically identified epithelial transporters and more than 130 uremic toxins [8]. However, the computational model safely assumed that the OAT1 was responsible for most IS transport on the basolateral membrane since the previous laboratory experiments measured a 7–10-fold decrease in IS binding affinity to other transporters such as OAT3 [7]. As such, the model does not account for toxin–toxin interactions or their competition for the same transporter and binding site. Third, the model also focused on the small-scale kinetics occurring in well plate volumes and excluded reacting species (albumin, IS, or OAT1) production and degradation. In order to upscale the model to replicate the BAK, the complexity of the model’s geometry would need to increase, including the flow and potential effects thereof on the transporter density, IS uptake, and dissociation rates. Finally, the laboratory experiments were performed in the absence of albumin, although there was evidence that albumin increases the clearance of IS in the bioartificial kidney (BAK) [6,7].

Additionally, the half-maximal inhibitory concentration (IC50) value of IS binding to OAT1 was reported in the presence of fluorescein and should be conducted with an incremental increase of IS to 500 µM. The uremic milieu and the large medication load that CKD patients were prescribed may lead to drug–toxin and toxin–toxin interactions, further complicating modeling the transporter functioning [21]. Therefore, the performance of the BAK, which relies on the excretory capacity of ciPTECs-OAT1, could be compromised in vivo. Understanding the drug–toxin and toxin–toxin interactions and the alterations in drug pharmacokinetics would be a step forward toward adequate polypharmacy adjustment in CKD patients. With these limitations in mind, we suggest that future work should focus on obtaining time-series experimental data of the uptake of IS in the presence of albumin, basolateral flow, and uremic plasma to extend the model proposed in this study.

## 4. Conclusions

To conclude, we have developed a computational model with an inward flux boundary condition that models the individual effects of (OAT1) transporter density, toxin (IS) uptake, and dissociation rate in the basolateral cell membrane that was fitted accurately and validated with the experimental results of IS uptake by a ciPTECs-OAT1 monolayer. The standard model allows investigation of the individual influences of the IS uptake rate ($kf_{Uptake})$, the dissociation rate of IS into the cell monolayer $(kf_{dissociation}),$ and the initial density of OAT1 on the basolateral cell membrane ($OAT1_{t=0}$). The standard model results suggested that IS removal was influenced mainly by the transporter density and IS uptake rate by the OAT1 and not by the initial albumin concentration as the model operated with excess albumin (1 mM). Additionally, the model was expanded to include albumin-binding effects in physiological and uremic conditions by altering the binding affinities of IS to albumin, resulting in the development of a uremic patient model including albumin conformational changes. In this scenario, the transporter density, IS uptake rate, and albumin binding became the most influential parameters on IS removal. By coupling computational and experimental data, the models developed within this study can be confidently considered to be a good representation of the IS transport processes in the proximal tubule by OAT1, within the limitations specified. Thus, the results of this study provide an exciting avenue to help understand the toxin–transporter intricacies in the PT and make better-informed decisions on BAK designs.

## 5. Materials and Methods

The computational models, i.e., a parameter-fitting model and the standard model, were developed in combination with the experimental setup described below. OAT1 was selected as the transporter of interest, considering the availability of a stable cell line that expresses the transporter in culture [16]. Indoxyl sulfate was selected as the uremic toxin of interest, since it is a protein-bound uremic toxin commonly transported by OAT1 and has detrimental effects in uremic patients [3,6,22]. The parameter-fitting model was a two-compartment model that was used to fit the unknown parameters of IS uptake rate by OAT1 from the well plate ($kf_{Uptake}),$ the dissociation rate of IS into the cell monolayer $\left(kf_{dissociation}\right)$, and the initial concentration of OAT1 on the basolateral cell membrane ($OAT1_{t=0}$), based on the IS experimental data (see Table 2). All parameter values and geometries used in the model are displayed in Table 2.

All computational modeling work was performed in the Virtual Cell Alpha 7.2.0 open-source systems biology platform developed at the University of Connecticut, Farmington, Connecticut [22].

### 5.1. Parameter-Fitting Model of Indoxyl Sulfate (IS) Transport

The parameter-fitting model of IS was developed to reflect the experimental time-series experiments of IS uptake in well plates (described in 5.7. *Experimental Work* below). The ‘well-mixed’ (uniform distribution of species) two-compartment model was developed with the basolateral cell membrane separating the well plate volume and cell cytoplasm (Figure 6a). The compartments were modeled with a volume of 100 µL (size blood) and a monolayer compartment volume of 3.2 × 10^8^ µm^3^ (size basolateral) with a cell membrane surface area of 0.32 cm^2^ (size membrane), corresponding to a standard 96-well plate setting (see Table 2 for parameter values). OAT1 was modeled using two-step mass action binding kinetics within the basolateral membrane to replicate IS uptake rate by the transporter ($kf_{Uptake})$ and the intracellular dissociation rate of IS $\left(kf_{dissociation}\right)$ (Figure 6a). These mass action binding kinetics resulted in the best fit of the experimental data (see Section “Selecting the $kf_{Uptake}$ function” below). The Breast Cancer Resistance Protein transporter (BCRP) efflux function was modeled as a Michaelis–Menten reaction within the apical membrane. BCRP is a known IS efflux transporter expressed on the apical membrane of the ciPTECs-OAT1 cell line [25,26]. We assume that the total amount of transporters and indoxyl sulfate was constant throughout the experiment (closed system). Moreover, since there was no albumin in the medium experimentally, we ignore the influence of albumin-IS binding in the parameter-fitting model to have a closer resemblance to the experimental setup. Equations (1)–(4) represent the system of ODEs describing the IS parameter-fitting model. The code of the IS parameter-fitting model can be found in the Virtual Cell database as “0D_IS_only”. The models were simulated to match the experimental sampling every 0.5 s within a timeframe of 45 min using Virtual Cell’s Combined Stiff Solver (IDA/CVODE) with a tolerance of 1 × 10^−9^.
(1)$$\frac{d\left[IS_{Well}\right]}{dt}=\left(kf_{\;Uptake}\left(IS_{Well}\times OAT1\right)+\frac{V_{Max,Efflux}IS_{Cell}}{K_{D,Efflux}+IS_{Cell}}\right)u$$
(2)$$\frac{d\left[OAT1_{IS}\right]}{dt}=\left(kf_{\;Uptake}\left(IS_{Well}\times OAT1\right)-kf_{dissociation}\left(IS_{OAT1}\right)\right)u$$
(3)$$\frac{d\left[OAT1\right]}{dt}=\left(-kf_{\;Uptake}\left(IS_{Well}\times OAT1\right)+kf_{dissociation}\left(IS_{OAT1}\right)\right)u$$
(4)$$\frac{d\left[IS_{Cell}\right]}{dt}=\left(kf_{dissociation}\left(IS_{OAT1}\right)-\frac{V_{Max,Efflux}IS_{Cell}}{K_{Efflux}+IS_{Cell}}\right)u$$
where ***u*** represents the conversion factor of the membrane reacting species to relate the cytosolic (volumetric) concentrations to the membrane concentrations:

conversion factor $u=\frac{size_{Membrane}}{size_{Well}}\cdot \frac{1}{N_{A}\times \mathrm{molecules}\times µM}$,

with Avogadro’ s constant $N_{A}=6.022\times 10^{23}\;{\mathrm{molecules}\;\mathrm{mole}}^{-1}$.

The unknown parameters in the system of equations ($kf_{Uptake}$ and $kf_{dissociation}$ and the total OAT1 transporter density $OAT1_{t=0}$) were fitted in the mathematical model using the experimental LC-MS/MS time-series data of intracellular concentration ($IS_{Cell}$) when $IS_{B,t=0}\;v$ = 25 µM measured at 1, 2, 5, 10, 15, 20, 25, 30, and 45 min (*n* = 6). The model was subsequently validated by the 25, 50, 100, and 500 µM LC-MS/MS data points at 1, 2, 5, 10, 15, 20, 25, 30, and 45 min (*n* = 6). The RMSE was used to quantify and compare how good the fit was for each function. In particular, the parameter-fitting model was solved for different values of $kf_{Uptake}$ and $kf_{dissociation}$ and the total OAT1 transporter density $OAT1_{t=0}$ to minimize the difference between the predicted and measured IS concentration inside the cell monolayer at seven different time points. For this, we used the *Evolutionary Programming* solver supported by COPASI (Virginia Tech, Blacksburg, VA, USA, http://copasi.org/, accessed on 18 September 2021). Evolutionary programming is a method to find the parameter values that result in the best fit of the experimental data and was inspired by the evolutionary theory of reproduction and selection. The first fitted values for the selected parameters (individual) asexually reproduce with one of the replicates undergoing a *mutation* or a slight alteration (competitor). The individual and the competitor compete, and the algorithm counts the number of times the individual outperforms the competitor. The individual results were ranked based on the number of wins, and the worst-fitted results were discarded, leaving the best fit/solution. The settings selected for COPASI are in Table 3, including the upper and lower limits of the fitting model. The parameter-fitting model was fitted three times with an increased number of runs to ensure the fitted parameter values ($kf_{Uptake}$ and $kf_{dissociation}$ and *OAT*1) had an error smaller than 1 × 10^−6^.

### 5.2. Selecting the $J_{Uptake}$ Function

Importantly, not only were the toxin transport parameters ($kf_{Uptake}$ and $kf_{dissociation}$) and the total OAT1 transporter density ($OAT1_{t=0}$) unknown, it was also unclear which type of kinetics describes the toxin transport best. In order to investigate this, we fitted the IS uptake rate by OAT1 ($kf_{Uptake}$) for various types of kinetics with increased complexity. More specifically, the parameter-fitting model was adjusted to have four cases of IS uptake ($J_{Uptake})$: (1) $kf_{Uptake}$ was a constant; (2) total membrane flux was lumped as Michaelis–Menten equations (3); and (4) there were two variations of the Michaelis–Menten equation. The COPASI parameter-fitting settings are specified in Appendix A.
(5)$$\mathrm{Case}\;1\;J_{Uptake}=kf_{Uptake}\left(IS_{Well}\times OAT1\right)$$
(6)$$\mathrm{Case}\;2\;J_{Uptake}=\frac{V_{Max}\times IS_{Well}}{K_{M}+IS_{Well}}$$
(7)$$\mathrm{Case}\;3\;J_{Uptake}=\frac{V_{Max}\times IS_{Well}}{K_{M}+IS_{Well}}\left(OAT1\right)$$
(8)$$\mathrm{Case}\;4\;J_{Uptake}=\frac{V_{Max}\times IS_{Well}}{K_{M}+IS_{Well}}\left(IS_{Well}\times OAT1\right)$$

The best $J_{Uptake}$ was selected based on the lowest root mean square error when compared among the four cases described in Equations (5)–(8) with $kf_{dissociation}$= 4.181 × 10^−4^ s^−1^ and $OAT1_{t=0}$= 1.15 × 10^7^ molecules µm^−2^.

### 5.3. Root Mean Square Error (RMSE)

The root mean square error was calculated, as shown in Equation (9), to evaluate the intracellular indoxyl sulfate (IS) best-fit curve with the experimental data. The RMSE was evaluated at 25, 50, 100, and 500 µM of IS’s initial concentration in the well plate at times 1, 2, 5, 10, 15, 20, 25, 30, and 45 min.
(9)$$\mathrm{RMSE}={\left[\frac{\sum_{i=1}^{n}{\left(IS_{Exp,i}-IS_{Sim,i}\right)}^{2}}{n}\right]}^{1/2}$$
where

$IS_{Exp,i}=$ concentration of IS at time *i* of the in vitro experiment,

$S_{Sim,i}=$ concentration of IS at time *i* of the simulation,

n = total number of data points,

i = data point corresponding to the time of measurement

### 5.4. Standard Model

For the standard model, we assumed similar kinetics as the parameter-fitting model with albumin. IS must initially bind to albumin to form a complex to interact with the transporter (OAT1). Once the IS–albumin complex binds to OAT1, albumin was returned to the blood compartment, and IS was transported into the cell (illustrated in Figure 6b). The standard model was considered to be a closed system with no generation of IS or OAT1. OAT1 was assumed to be evenly distributed along the membrane length. The standard model was simplified to an ODE system with seven reacting species to simulate a boundary influx with a two-step IS binding with the OAT1 transporter at the basolateral cell membrane and a Michaelis–Menten efflux boundary condition on the apical cell membrane to represent IS transport by the BCRP into the apical compartment (see Figure 6b). The initial conditions and parameter settings, determined from the literature and the parameter fitting model, are represented in Table 2. The model was simulated using Equations (10)–(16).

Similar to the experimental and fitting setup, the standard model has the same compartment geometries and thus volumes as the fitting model to reduce size effects on the simulation, resulting in three compartments: blood (100 µL), cell monolayer (3.2 × 10^8^ µm^3^), and dialysate (100 µL). Both basolateral and apical membranes (0.32 cm^2^) were modeled as interfaces between the blood and cell compartments and the cell and dialysate compartments, respectively.

The standard model was simulated until there was complete removal of indoxyl sulfate (17.5 h) using a Virtual Cell’s Combined Stiff Solver (IDA/CVODE) with a tolerance of 1 × 10^−9^, sampling every 50 s. The standard model used the fitted parameters, $kf_{Uptake}$ and $kf_{dissociation}$ and OAT1 from the IS parameter fitting model. The standard model code can be found in the Virtual Cell database as “0D_Full_3Compartment_ IS” (University of Connecticut, Farmington, Connecticut [22]).

Note that we modeled various albumin conditions ranging from scenarios with 1 mM albumin (standard model—Section 5.4, physiological condition and uremic condition—Section 5.6) or without albumin (for the parameter-fitting model, since the experiments were performed in the absence of albumin—Section 5.1).
(10)$$\frac{d\left[IS_{B}\right]}{dt}=-Kf_{Complex}\left(IS_{B}\right)$$
(11)$$\frac{d\left[HSA\right]}{dt}=-Kf_{Complex}\left(IS_{B}\right)+\left(kf_{\;Uptake}\left(Complex\times OAT1\right)\right)u_{B-B}$$
(12)$$\frac{d\left[Complex\right]}{dt}=Kf_{Complex}\left(HSA\times IS_{B}\right)-\left(kf_{Uptake}\left(Complex\times OAT1\right)\right)u_{B-B}$$
(13)$$\frac{d\left[OAT1\right]}{dt}=\left(-kf_{Uptake}\left(Complex\times OAT1\right)+kf_{dissociation}\left(IS_{OAT1}\right)\right)u_{B-B}$$
(14)$$\frac{d\left[OAT1_{IS}\right]}{dt}=\left(kf_{\;Uptake}\left(Complex\times OAT1\right)\right)u_{B-B}-\left(kf_{dissociation}\left(IS_{OAT1}\right)\right)u_{B-C}$$
(15)$$\frac{d\left[IS_{Cell}\right]}{dt}=\left(kf_{dissociation}\left(IS_{OAT1}\right)\right)u_{B-C}-\frac{V_{Max,Efflux}\times IS_{Cell}}{K_{D,Efflux}+IS_{Cell}}u_{A-C}$$
(16)$$\frac{d\left[IS_{D}\right]}{dt}=\frac{V_{Max,Efflux}\times \left[IS_{Cell}\right]}{K_{D,Efflux}+[IS_{Cell}]}u_{A-D}$$
where,



$Kf_{Complex}=\frac{V_{Max,HSA}\times \left[HSA\right]}{K_{D,HSA}+\left[HSA\right]}$





$u_{B-B}=\frac{size_{Basolateral}}{size_{Blood}}\cdot \frac{1}{N_{A}\times \mathrm{molecules}\times µM}$





$u_{B-C}=\frac{size_{Basolateral}}{size_{Cell}}\cdot \frac{1}{N_{A}\times \mathrm{molecules}\times µM}$





$u_{A-D}=\frac{size_{Apical}}{size_{Dialysate}}\cdot \frac{1}{N_{A}\times \mathrm{molecules}\times µM}$





$N_{A}=6.022\times 10^{23}\;{\mathrm{molecules}\;\mathrm{mole}}^{-1}$



### 5.5. Sensitivity Analysis

A sensitivity analysis was performed on the standard model where the individual parameter values were altered to investigate which parameter was most influential on the IS concentration in the dialysate ($IS_{D}$). The sensitivity analysis was performed for the transporter density ($OAT1_{t=0}$), uptake ($kf_{IS,\;Uptake}$) and dissociation rate ($kf_{dissociation}$), albumin concentration (HSA,) and albumin-binding rate ($Kf_{Complex}$) to IS using Equation (17). A total of 17.5 h was chosen as the end time point for the standard and uremic models as it was the time taken for the complete removal of IS in the standard model.
(17)$$Sensitivity=\frac{\left|IS_{D}\left(k\right)-IS_{D}\left(k+\Delta k\right)\right|}{IS_{D}\left(\Delta k\right)}/\frac{\Delta k}{k}$$
where

$IS_{D}\left(k\right)$ = IS in the dialysate using the standard model at 17.5 h

$IS_{D}\left(k+\Delta k\right)=$ IS in the dialysate at + or − 20% of the standard model parameter values at 17.5 h

Δk= varied parameter

k= standard model parameter value

### 5.6. Developing a Uremic Model

The standard model was adapted to account for physiological ($IS_{t=0}=2.5\;µM$) and uremic ($IS_{t=0}=180\;µM)$ concentrations. The effect of albumin conformational changes was included in the uremic condition model to investigate the uremic patient condition further. The binding coefficients of IS to albumin ($Kf_{Complex})$ and uptake rate ($kf_{Uptake})$ were multiplied by a reducing factor f1 and f2, respectively, to mimic these albumin conformational changes in uremic conditions. A parameter scan was performed to find the best parameter set of f1 and f2 to result in an 85% reduction of IS [7] removal when compared to the model with no conformational changes at 17.5 h.

### 5.7. Experimental Work

#### 5.7.1. Cell Culture of CiPTECs-OAT1

Conditionally immortalized proximal tubule epithelial cells obtained from urine samples of healthy volunteers and overexpressing the organic anion transporter 1 (ciPTECs-OAT1) were cultured as described by Nieskens et al. [25]. Cells were seeded at a density of 63,000 cells/cm^2^, cultured at 33 °C to allow expansion, then cultured for 7 days at 37 °C for differentiation and maturation. Cells were cultured using Dulbecco’s modified eagle medium (DMEM HAM’s F12, Life Technologies, Paisly, UK), 5 μg/mL insulin, 5 μg/mL transferrin, 5 μg/mL selenium, 35 ng/mL hydrocortisone, 10 ng/mL epidermal growth factor (EGF), 40 pg/mL tri-iodothyronine (Sigma-Aldrich, Zwijndrecht, The Netherlands), and 10% fetal calf serum (FCS, Greiner Bio One, Kremsmuenster, Austria).

#### 5.7.2. Exposure to IS

To determine the time-dependent OAT1-mediated intracellular uptake of IS, mature monolayers of ciPTECs-OAT1 were incubated with IS (25, 50, 100, and 500 μM prepared in Krebs–Henseleit buffer (Sigma-Aldrich, Zwijndrecht, The Netherlands) supplemented with HEPES (10 mM, Sigma-Aldrich, Zwijndrecht, The Netherlands, pH 7.4) for variable periods of time. Uptake was stopped by washing one time with ice-cold HBSS (Life Technologies Europe BV, Roskilde, Denmark), and then, the cells were lysed by 100 µL 0.1M NaOH for 10 min at room temperature and under mild shaking. The intracellular IS concentration was determined in the cell lysate by LC-MS/MS, as described below.

#### 5.7.3. Intracellular Detection of IS

##### Reagents

IS potassium salt and isotope-labeled IS potassium salt (13C6, 99%) as internal standard were purchased from Sigma-Aldrich (Zwijndrecht, The Netherlands) and Cambridge Isotope Laboratory (Tewksbury, Massachusetts, USA), respectively. Water (U°LC-MS grade), acetonitrile (HPLC-S grade), and methanol (HPLC grade) were purchased from Biosolve (Valkenswaard, The Netherlands). Formic acid (analytical grade) was obtained from Merck (Darmstadt, Germany). Ultrapure water was produced by a Milli-Q^®^ Advantage A10 Water Purification Systems (Merck, Amsterdam, The Netherlands).

##### Equipment

The LC-MS/MS system consisted of a DGU-14A degasser, a CTO-10Avp column oven, a Sil-HTc autosampler, and two LC10-ADvp- pumps (Shimadzu, Kyoto, Japan) and a Finnigan TSQ Quantum Discovery Max triple quadrupole mass spectrometer with electrospray ionization (Thermo Electron, Waltham, MA, USA). The Xcalibur software (version 1.4, Thermo Electron) was used to record and process the data.

##### Sample Pre-Treatment

The cell lysate underwent the protein precipitation procedure. About 20 μL of cell lysate was crushed by 80 μL of protein precipitant containing 0.5 µg/mL of internal standard, which was followed by vortexing for 2 min and centrifuging for 2 min at 1000 rpm. After that, 64 µL of supernatant was collected in 1 mL round-bottom wells of a polypropylene 96-deep well plate and diluted with 200 µL of ultrapure water. Finally, the plate was gently shaken before placing it in the autosampler for LC-MS/MS analysis.

## Figures and Tables

**Figure 1 toxins-13-00674-f001:**
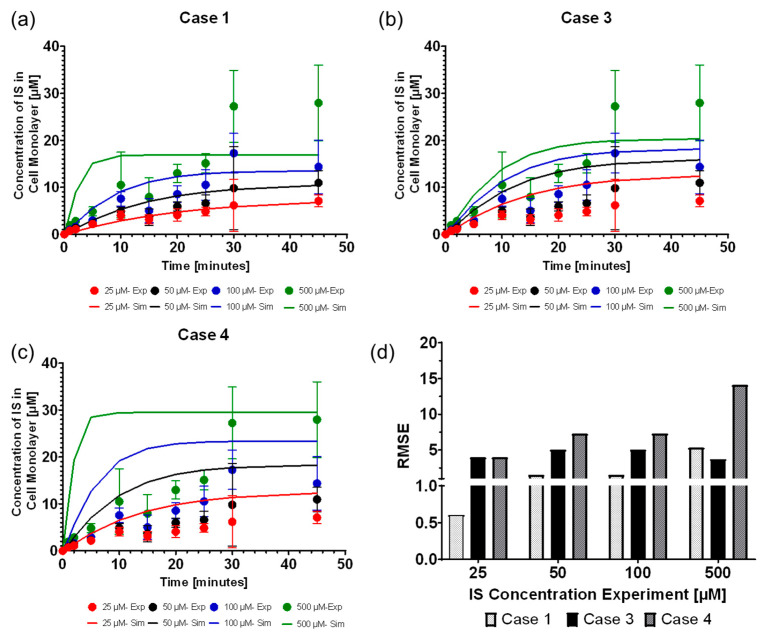
Selecting the best fit function of the IS uptake boundary condition. The parameter-fitting model was fitted with the $IS_{B,t=0}$= 25 µM and validated with $IS_{t=0}$ = 50, 100, and 500 µM. The continuous line represents the simulated data, and the dots with standard deviations (*n* = 6) represent the LC-MS/MS IS data (**a**) Case 1: Fitting results based on $k_{f,uptake}$. (**b**) Case 3: fitting results based on $J_{Uptake}=\frac{V_{Max}\times IS_{Well}}{K_{M}+IS_{Well}}\left(OAT1\right)$. (**c**) Case 4: fitting results based on $J_{Uptake}=\frac{V_{Max}\times IS_{Well}}{K_{M}+IS_{Well}}\left(IS_{B}\times OAT1\right)$. (**d**) Comparison of RMSE for cases 1, 3, and 4.

**Figure 2 toxins-13-00674-f002:**
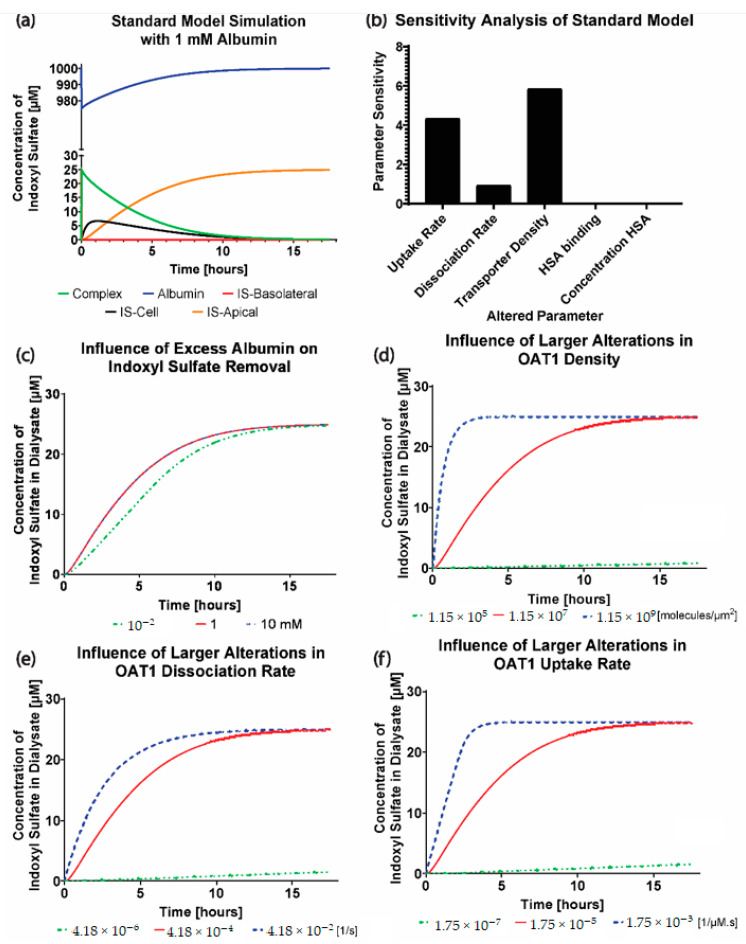
Simulations of the standard model using the fitted parameters of the parameter-fitting model. (**a**) A 17.5 h extended simulation of the standard model to show the trends of 25 µM indoxyl sulfate (IS) complexed with albumin, unbound IS in the cell monolayer, and unbound IS in the dialysate. (**b**) Sensitivity analysis of the standard model to investigate the influence of a 20% decrease in transporter density, uptake and dissociation rate, albumin concentration, and albumin binding rate to IS on IS transport to the dialysate. (**c**) Step variations of initial albumin concentration (1 × 10^−2^, 1 × 10^0^ and 1 × 10^2^ mM) in the blood compartment. (**d**) Step variations of unbound OAT1 density (1.15 × 10^5^, 1.15 × 10^7^, 1.15 × 10^9^ molecules/µm^2^) along the basolateral cell membrane. (**e**) Step variations to the dissociation rate (4.18 × 10^−6^, 4.18 × 10^−4^, 4.18 × 10^−2^ s^−1^) of IS from OAT1 into the cell monolayer. (**f**) Step variations to the uptake rate of IS (1.75 × 10^−7^, 1.75 × 10^−5^, 1.75 × 10^−3^ s^−1^ µM^−1^) by OAT1 in the blood compartment.

**Figure 3 toxins-13-00674-f003:**
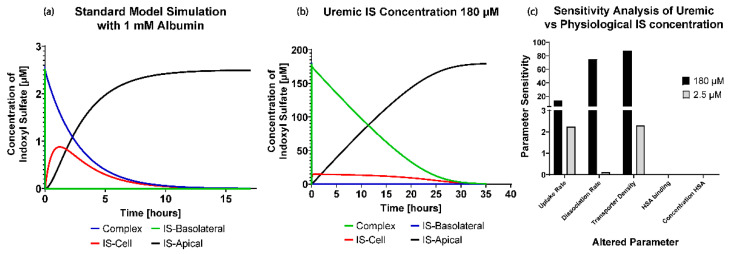
(**a**) Full dynamic profile of physiological IS concentration with 17.5 h of simulation time. (**b**) Full dynamic profile of uremic IS concentration with 35 h of simulation time. (**c**) Sensitivity analysis to investigate the influence of a 20% decrease of the standard model parameters for physiological and uremic conditions of indoxyl sulfate in uremic conditions.

**Figure 4 toxins-13-00674-f004:**
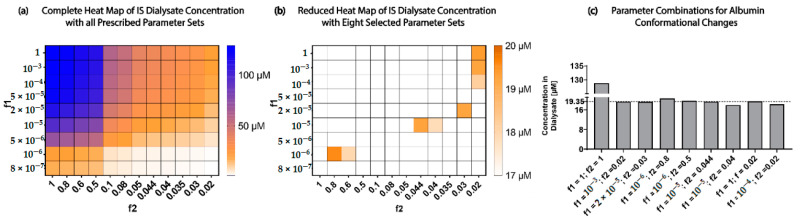
Albumin conformational changes parameter selection using parameter scan of f1 and f2. (**a**) Complete heat map using the $IS_{dialysate}$ concentration of all parameter sets of f1 and f2 ranging from 130 µM (blue) to 0 µM (white). (**b**) A reduced heat map with a narrowed concentration range of 17 to 20 µM to select the six best parameter sets of f1 and f2. (**c**) Bar graph for the selected eight parameter sets that result in an $IS_{D}$ concentration closest to 19.35 µM (dotted line).

**Figure 5 toxins-13-00674-f005:**
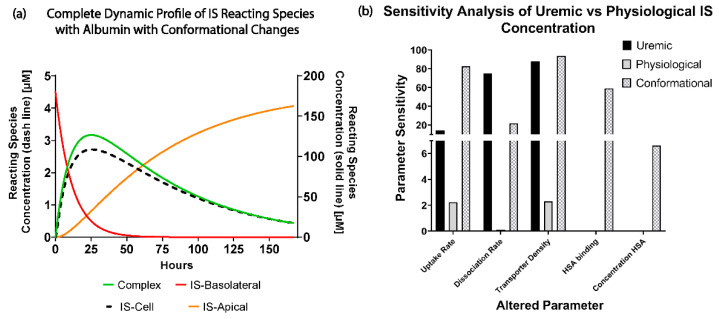
Simulations of the uremic conditions using *I*$S_{B,t=0}=180\;µM;$ f1 = 1 × 10^−5^ and f2 = 0.0.44. (**a**) The complete dynamic profile of the IS reacting species in the simulations until complete removal (84 h time period). The right axis plots intracellular IS (IS-Cell) concentration. While the left axis plots the IS–albumin complex (Complex) and IS concentration in the basolateral and apical compartments. (**b**) Sensitivity analysis comparing the uremic ($IS_{B,t=0}$ = 180 µM) and physiological ($IS_{B,t=0}$ = 2.5 µM) IS concentrations with albumin conformational changes (f1 = 1 × 10^−5^, f2 = 0.044 and $IS_{B,t=0}$ = 180 µM) by reducing the individual parameters (uptake rate, dissociation rate, transporter density, albumin binding, and albumin concentration) by 20%.

**Figure 6 toxins-13-00674-f006:**
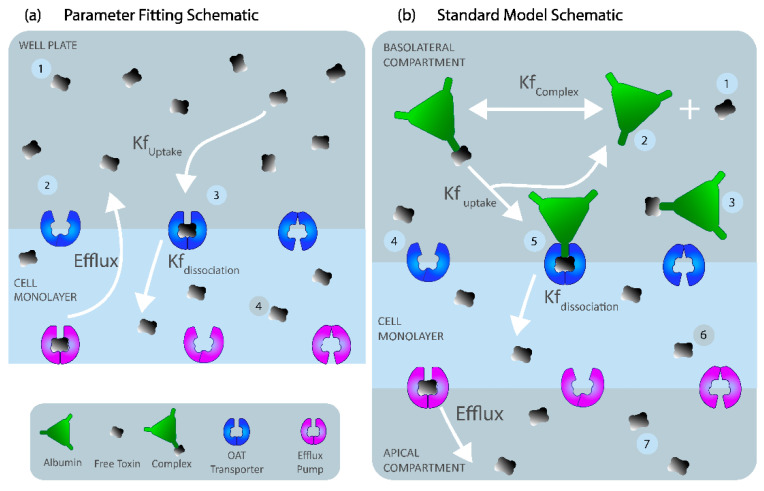
(**a**) Schematic of the parameter-fitting model based on data from IS cellular uptake experiments with four reacting species: (1) free indoxyl sulfate, $IS_{Well}$; (2) unbound OAT1; (3) IS bound to OAT1, $IS_{OAT1}$; (4) IS transported to the proximal tubule monolayer, $IS_{Cell}$. IS binds to OAT1 on the ciPTECs-OAT1 basolateral membrane at a rate $kf_{IS,Uptake}$ and dissociates from the cell monolayer at a rate $kf_{Dissociation}$. The IS is transported back to the well at the efflux rate of the breast cancer-resistant protein (BCRP) efflux pump. (**b**) Schematic of the standard protein-bound uremic toxin model. The model uses seven reacting species: (1) free indoxyl sulfate, $IS_{B}$; (2) human serum albumin; (3) IS bound to albumin, Complex; (4) unbound OAT1; (5) IS bound to OAT1, $IS_{OAT1}$; (6) IS transported to the proximal tubule monolayer, $IS_{Cell}$; (7) IS excreted to the dialysate/lumen, $IS_{D}$. The reactions occur within three compartments: the blood, the proximal tubule epithelial cell monolayer, and the dialysate. The basolateral membrane separates the blood and the cell monolayer, whereas the apical membrane separates the cell monolayer and the dialysate.

**Table 1 toxins-13-00674-t001:** Table of the fitted parameters after using 25 µM indoxyl sulfate in the parameter-fitting model.

Parameter	Value		Unit
$kf_{Uptake}$	Case 1	1.75 × 10^−5^	s^−1^ µM^−1^
	Case 2	Unable to fit	
	Case 3	*V_max_* = 1.01 × 10^−3^ s^−1^; *Km* = 1.99 × 10^2^ µM	
	Case 4	*V_max_* = 4.47 × 10^−6^ s^−1^; *K_m_* = 4.06 × 10^−5^ µM	
$kf_{dissociation}$	4.18 × 10^−4^		s^−1^
$OAT1_{t=0}$	1.15 × 10^7^		transporters µm^−2^

**Table 2 toxins-13-00674-t002:** Main parameters for the parameter fitting and standard Models. Note that we modeled various albumin condiions in mM albumin (standard model—Section 5.4, physiological condition and uremic condition—Section 5.6) or without albumin (for the parameter-fitting model since the experiments were performed in the absence of albumin—Section 5.1).

Name of Species	Parameter Symbol in Model	Value		References
		Parameter Fitting Model	Standard Model	
Initial albumin [mM]	$HSA_{t=0}$	0	1	[6,7]
Initial indoxyl sulfate [µM]	$IS_{t=0}$	$IS_{Well}=$25, 50, 100, 500	$IS_{B}=$2.5 and 180	Experimental conditions
Initial complex [µM]	$Complex_{t=0}$	0	0	-
Initial IS bound OAT1 [molecules µm^−2^]	$IS_{OAT1,t=0}$	0	0	-
Initial indoxyl sulfate in Cell [µM]	$IS_{Cell,t=0}$	0	0	-
Initial indoxyl sulfate in Dialysate [µM]	$IS_{D,t=0}$	-	0	-
Initial OAT1 [molecules µm^−2^]	$OAT1_{t=0}$	1.15 × 10^7^		Fitted
Uptake by OAT1 [s^−1^ µM^−1^]	$kf_{\;Uptake}$	1.75 × 10^−5^		Fitted
Dissociation from OAT1 [s^−1^]	$kf_{Dissociation}$	4.18 × 10^−4^		Fitted
Efflux [molecules µm^2^ s^−1^]	$J_{\mathrm{Efflux}}$	$\frac{V_{Max,Efflux}\times IS_{Cell}}{K_{Efflux}+IS_{Cell}}$		[23]
	$V_{E,Max}$ $\left[{\mathrm{molecules}\;µm}^{2}{\;s}^{-1}\right]$	24,000		
	$K_{Efflux}$ [µM]	69		
Kf Binding of IS to albumin [s^−1^ µM^−1^]	$Kf_{Complex}$	$\frac{B_{Max}\times \left[HSA\right]}{K_{D}+\left[HSA\right]}$		[24]
	$B_{Max}$ [s^−1^]	2.70		
	$K_{D}$ [µM]	97.92		
Volume of 96-Well [µL]	*Size well*	100		Thermo Scientific
Area of Basolateral Cell Membrane [cm^2^]	*Size membrane*	0.32		Thermo Scientific
Volume of Monolayer [µm^3^]	*Size cell*	3.2 × 10^8^		Thermo Scientific

**Table 3 toxins-13-00674-t003:** COPASI parameter fitting settings for the evolutionary programming solver chosen to fit the parameter fitting model.

COPASI Parameter Fitting Settings for the Evolutionary Programming Solver			
Number of generations	200	Number of generations the population evolves	
Population size	20	Number of individuals that survive	
Seed	1	Random number generator	
Number of runs	1, 5, 10	Increased number of runs to check if increasing the number of runs alters the fitted value	
**COPASI Parameter Fitting Settings for Guessing Fitted Values for the Parameter Fitting Model**			
**Parameter**	**Initial Guess**	**Lower Limit**	**Upper Limit**
$kf_{Uptake}$ [s^−1^ µM^−1^] IS binding to transporter	1 × 10^−4^	5 × 10^−7^	1 × 10^−3^
$kf_{dissociation}$ [s^−1^] IS dissociation from transporter	1 × 10^−3^	5 × 10^−5^	1 × 10^−3^
[OAT1] [molecules µm^−2^] Density of transporter	5 × 10^6^	7 × 10^5^	2 × 10^7^

## Data Availability

All models are made public in the Virtual Cell database.

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
