# Peer review of "The Influence of OAT1 Density and Functionality on Indoxyl Sulfate Transport in the Human Proximal Tubule: An Integrated Computational and In Vitro Study"

_toxins, 2021, doi:10.3390/toxins13100674_

Round 1
Reviewer 1 Report
I apologize for not complying with the deadline, but I had to read the manuscript repeatedly, because after the first reading it was not at all clear to me what this article is about and which are actually mathematical and which experimental results. The experimental work states that the LC-MS / MS system is used, but nowhere in the work did I notice the results of measurements or any comparisons.
It is clear that the authors have done a good job, but since the methodology states that it is a mathematical-experimental model, it would be appropriate to visibly compare these results in the abstract, results and in the discussion and conclusion. The reader is missing ...
The results are incomprehensible to me and the article needs to be read repeatedly so that the reader understands some context. They may be understandable to narrow-minded scientists, but if this article is to be useful to the larger circle, some minor adjustments are required.
Also check the abbreviation RMSE, because in some places you mention RSME (p.3)
And I'm also confused with the effect of albumin ... on page 4 you state that it has no effect on transport and then on page 5 you describe that after reducing the concentration of albumin, the IS removal from the system is also reduced .......... On page 6 in fig.3 you state that albumin does not affect the removal of IS ....... and in the end I was quite confused when on page 9 you state that the experiments were done in the absence of albumin.
In conclusion, I would like to point out once again that the results are nice, but it is necessary to visibly explain to the reader what is the calculation and what is the experiment.
Author Response
Response to Reviewer 1:
We thank the reviewer for their comments and apologize for any confusion in the manuscript, we hope that the following changes address these concerns satisfactorily.
I apologize for not complying with the deadline, but I had to read the manuscript repeatedly, because after the first reading it was not at all clear to me what this article is about and which are actually mathematical and which experimental results. The experimental work states that the LC-MS / MS system is used, but nowhere in the work did I notice the results of measurements or any comparisons.
It is clear that the authors have done a good job, but since the methodology states that it is a mathematical-experimental model, it would be appropriate to visibly compare these results in the abstract, results and in the discussion and conclusion. The reader is missing ...
In conclusion, I would like to point out once again that the results are nice, but it is necessary to visibly explain to the reader what is the calculation and what is the experiment.
The results of the LC-MS/MS were compared with the simulation results in figure 1. In the original manuscript, we specify this in the results sections 2.1. Fitted Results Using 25 µM Indoxyl Sulfate (IS) and 2.1.1. Selection of IS uptake boundary condition function. However, to clarify this in the methods section we added the following statement to 5.1. Parameter Fitting Model of Indoxyl Sulfate (IS) Transport:
“The unknown parameters in the system of equations (kf_Uptake and 〖kf〗_dissociation and the total OAT1 transporter density 〖OAT1〗_(t=0)) were fitted in the mathematical model using the experimental LC-MS/MS time-series data of intracellular concentration (IS_Cell) when IS_(B,t=0)= 25 µM measured at 1, 2, 5, 10, 15, 20, 25, 30, and 45 minutes (n=6) . The model was subsequently validated by the 25, 50, 100, and 500 µM LC-MS/MS data points at 1, 2, 5, 10, 15, 20, 25, 30, and 45 minutes (n=6). The RMSE was used to quantify and compare how good the fit was for each function.”
And adapted figure 1’s caption to read as follows “
“Figure 1. Selecting the best fit function of the IS uptake boundary condition. The Parameter Fitting Model was fitted with the IS_(B,t=0)= 25 µM and validated with IS_(t=0)= 50, 100, and 500 µM. The continuous line represents the simulated data, and the dots with standard deviations (n = 6) represent the LC-MS/MS IS data (a) Case 1: Fitting results based on k_(f,uptake). (b) Case 3: fitting results based on J_Uptake=(V_Max*IS_Well)/(K_M+IS_Well ) (OAT1). (c) Case 4: fitting results based on J_Uptake=(V_Max*IS_Well)/(K_M+IS_Well ) (IS_B×OAT1). (d) Comparison of RMSE for cases 1, 3, and 4.”
Also check the abbreviation RMSE, because in some places you mention RSME (p.3)
We would like to thank the reviewer for bringing this typo to our attention. All the instances of “RSME” in the manuscript have been corrected.
And I'm also confused with the effect of albumin ... on page 4 you state that it has no effect on transport and then on page 5 you describe that after reducing the concentration of albumin, the IS removal from the system is also reduced .......... On page 6 in fig.3 you state that albumin does not affect the removal of IS ....... and in the end I was quite confused when on page 9 you state that the experiments were done in the absence of albumin.
We would like to apologize for this confusion in regards to albumin in the manuscript. There was a historical text error on page 5 in the following sentence which has now been corrected:
“Likewise, decreasing the albumin concentration to 1E-2 mM, results in a small reduction in the rate of IS removal from the system”.
In regards to page 9, we model various albumin-conditions ranging from scenarios with with 1 mM albumin (physiological condition and uremic condition) or without albumin (for the parameter fitting model since the experiments are performed in the absence of albumin). Figure 3 is a simulation scenario that involves albumin binding assuming the interaction of albumin and IS as measured in previous studies. To help the reader, we have added which section in the methods corresponds to each model results’ section and the follow text to the methods section:
“Note that we modeled various albumin conditions ranging from scenarios with 1 mM albumin (standard model-section 5.4, physiological condition and uremic condition – section 5.6) or without albumin (for the parameter fitting model since the experiments were performed in the absence of albumin – section 5.1).”
And
“Table 2. Main parameters for the Parameter Fitting and Standard Model. Note that we modeled various albumin conditions ranging from scenarios with 1 mM albumin (standard model-section 5.4, physiological condition and uremic condition – section 5.6) or without albumin (for the parameter fitting model since the experiments were performed in the absence of albumin – section 5.1).”
In addition, the limitation concerning the absence of albumin refers to the laboratory experiments, so to make this clear we have adjusted the text as:
“ Finally, the laboratory experiments were performed in the absence of albumin, although there is evidence that albumin increases the clearance of IS in the bioartificial kidney (BAK)[3,4]”

Reviewer 2 Report
The current manuscript developed a proximal tubule computational model focusing on the organic anionic transporter 1 (OAT1) mediated indoxyl sulfate transport by including the transporter density, the activity of the transporter, dissociation rate as well as albumin interaction. The study was well organized and below are the detailed comments:
1. Indoxyl sulfate was selected as the uremic toxin of interest since it is a protein-bound uremic toxin commonly transported by ???1. How about the role of OAT3 in the tranport of indoxyl sulfate?
2. During the selection of the best fit function of the indoxyl sulfate uptake boundary condition (Fig. 1), different equations can not fit well for the highest concentration (500 uM) group data. Why?
3. Standard model simulation results demonstrated that OAT1 density played a critical role in transport of indoxyl sulfate. How was the OAT1 density determined for specific model simulation including such paramenter?
Author Response
Reponse to Reviewer 2:
Indoxyl sulfate was selected as the uremic toxin of interest since it is a protein-bound uremic toxin commonly transported by ???1. How about the role of OAT3 in the tranport of indoxyl sulfate?
We would like to thank the reviewer for this comment about the potential role of OAT3 on IS transport. The laboratory experiments were conducted with a ciPTEC-OAT1 cell line, where the transport is assumed to be dominated by the OAT1 as the IS binding affinity to OAT3 was previously measured to be 7-10 fold lower when compared to OAT1[4]. This justifies the assumption that OAT1 is the essential transporter for IS in these modeling studies. To investigate the effect of OAT3, similar experiments would need to be done with a ciPTEC-OAT3 cell line. To make this assumption clear we have added the following statement in the discussion as a limitation:
“However, the computational model safely assumed that the OAT1 is responsible for most IS transport across the basolateral membrane since previous laboratory experiments measured a 7-10 fold decrease in IS binding affinity to other transporters such as OAT3 [4].”
During the selection of the best fit function of the indoxyl sulfate uptake boundary condition (Fig. 1), different equations can not fit well for the highest concentration (500 uM) group data. Why?
We would like to thank the reviewer for this comment. The difficulty of fitting 500 µM is likely due to the saturation of the OAT1 by the high IS concentration levels. We have addressed the issue in the fitting in the original manuscript in lines 129-135 as shown below:
“Figure 1a shows that the fitted kf_Uptake as a constant (case 1) was the most optimal fit resulting in the lowest RMSE for each time point (Figure 1d), closely followed by case 3 (Figure 1b). All cases, particularly case 4 (Figure 1c) had difficulty fitting 500 µM, giving rise to the largest RMSE. The difficulty of fitting 500 µM was likely due to the oversaturation of the OAT1 by the high IS concentration levels. IS binding with OAT1 has an IC50 value of 25 ± 4 µM with fluorescein [3], alluding to higher IS concentrations inhibiting OAT1.”
Standard model simulation results demonstrated that OAT1 density played a critical role in transport of indoxyl sulfate. How was the OAT1 density determined for specific model simulation including such parameter?
We thank the reviewer for this pertinent comment. The OAT1 density was determined through the fitting of dedicated, experimental time-series data. We apologize that this did not come across clearly in the manuscript. In the original manuscript, we describe the fitting of the transporter density in the methods section 5.1. Parameter Fitting Model of Indoxyl Sulfate (IS) Transport and Table 3. We have adapted the text to clarify:
“The Standard Model allows investigation of the individual influence of the IS uptake rate by OAT1 (kf_Uptake) and the dissociation rate of IS into the cell monolayer 〖(kf〗_dissociation) and the initial concentration of OAT1 on the basolateral cell membrane (OAT1_(t=0)). With the Parameter Fitting Model calibrated and validated with the experimental time-series data, we used the calibrated model parameters (kf_Uptake=1.75E-5 s^(-1).µM^(-1),〖kf〗_dissociation=4.18E-4 s^(-1) and OAT1=1.15E7 trasnporters.µm^(-2), see the results of Table 1: Case 1) in the Standard Model to explore the influence of the presence of 1 mM albumin, similar to other ciPTECs-OAT1 experiments[3,4].”

Reviewer 3 Report
Dear Authors,
Thank you for the opportunity to review your draft entitled The Influence of OAT1 Density and Functionality on Indoxyl Sulfate Transport in the Human Proximal Tubule: an Integrated Computational and In Vitro Study.
I found the presented manuscript well-written, interesting to the Readers as well as discussing important medical issues in the field of nephrology.
I have only some minor points that should be addressed during the review process:
- Please expand keywords - they do not contain anything related to IS that might be misleading
- In the introduction section, it would be beneficial to add one paragraph describing the toxicity of IS towards the human body - it will underlie the importance of your finding and will make readers more familiar with IS biology. Below pasted papers will be handy - they are comprehensive and well written.: 1) https://doi.org/10.1186/s12882-017-0457-1 2) https://doi.org/10.1159/000368488 3) https://doi.org/10.2147/IJNRD.S287237
- What was the reason for choosing these concentrations of IS? Is there any literature reference considering animal/human models?
- Did the Authors consider other OATs as valuable in IS biology research?
Moreover, the discussion section is well written and the Authors fairly discussed the potential limitation of the study - (line 310) what make the paper suitable for the publication in the Journal after implemented minor changes as stated above.
Author Response
Reponse to Reviewer 3:
Please expand keywords - they do not contain anything related to IS that might be misleading.
We would like to thank the review and propose the following new keywords:
“Indoxyl Sulfate; Organic Anionic Transporter; Uremic Toxins; Albumin Conformational Changes; Kinetic Modeling”
In the introduction section, it would be beneficial to add one paragraph describing the toxicity of IS towards the human body - it will underlie the importance of your finding and will make readers more familiar with IS biology. Below pasted papers will be handy - they are comprehensive and well written.: 1) https://doi.org/10.1186/s12882-017-0457-1 2) https://doi.org/10.1159/000368488 3) https://doi.org/10.2147/IJNRD.S287237
We thank the reviewer for pointing out the gap in the introduction. We have added the following line to bring focus to the complications of IS and other uremic toxins.
“However, if these uremic toxins remain in the bloodstream, the patient will suffer from a number of complications including, but not limited to, cardiovascular disease, anemia and progressive kidney failure [3–5]”
What was the reason for choosing these concentrations of IS? Is there any literature reference considering animal/human models?-
We would like to thank the reviewer for this pertinent comment. We selected the IS concentration based on the physiological and uremic state according to the Eutox database (https://database.uremic-toxins.org/soluteDetails.php?solute_id=178). We would like to direct the reviewer to the results (section- 2.3) where we have clarified this decision:
“According to the EUTox database, the plasma levels of IS measured in a patient are around 2.5 µM (~0.54 µg/mL) in physiological conditions and 180 µM (~37.07 µg/mL) in uremic conditions. Applying these initial conditions to the Standard Model parameters (Table 1: Case 1) resulted in the dynamic profiles shown in Figure 4a-b.”
Did the Authors consider other OATs as valuable in IS biology research?
We would like to thank the reviewer for their comment regarding other OATs. We acknowledge that other OATs are critical for toxin transport, such as OAT3 for example (as also mentioned by reviewer 2). However, this study focuses on OAT1, since the laboratory experiments were conducted with a ciPTEC-OAT1 cell line. More specifically, previous measurements have indicated that the IS binding affinity to OAT3 was 7-10 fold lower when compared to OAT1, indicating that the transport is dominated by OAT1 [4]. This justifies the assumption that OAT1 is the essential transporter for IS in these modeling studies. We included the following statement in the discussion to substantiate our assumption:
“However, the computational model safely assumed that the OAT1 is responsible for most IS transport across the basolateral membrane since previous laboratory experiments measured a 7-10 fold decrease in IS binding affinity to other transporters such as OAT3 [4].”
